# Identification of Potential Host Plants of Sap-Sucking Insects (Hemiptera: Cicadellidae) Using Anchored Hybrid By-Catch Data

**DOI:** 10.3390/insects12110964

**Published:** 2021-10-23

**Authors:** Yanghui Cao, Christopher H. Dietrich

**Affiliations:** Illinois Natural History Survey, Prairie Research Institute, University of Illinois, Champaign, IL 61820, USA; chdietri@illinois.edu

**Keywords:** leafhoppers, host plants, insect–plant association, anchored hybrid enrichment

## Abstract

**Simple Summary:**

Plant–insect interactions are a significant driver of terrestrial biodiversity. Evolutionary studies of such interactions have been hindered by the lack of reliable host plant records, which have primarily been obtained through field observations. More recently, traditional or next-generation sequencing methods have been used successfully to detect host plant DNA in DNA extracted from plant-feeding insects, but most such studies have focused on chewing insects that ingest plant tissues with large quantities of DNA. In this study, next-generation sequencing data were used to determine the feasibility of detecting plant genes in sap-sucking insects, which may ingest very little plant DNA. Although no plant-specific probes were used to generate the sequence data, multiple plant genes were retrieved in the by-catch data. Our results suggest that next-generation sequencing may present a powerful tool for detecting and characterizing potential host plants using DNA extracted from phytophagous insects, including sap feeders.

**Abstract:**

Reliable host plant records are available for only a small fraction of herbivorous insect species, despite their potential agricultural importance. Most available data on insect–plant associations have been obtained through field observations of occurrences of insects on particular plants. Molecular methods have more recently been used to identify potential host plants using DNA extracted from insects, but most prior studies using these methods have focused on chewing insects that ingest tissues expected to contain large quantities of plant DNA. Screening of Illumina data obtained from sap feeders of the hemipteran family Cicadellidae (leafhoppers) using anchored hybrid enrichment indicates that, despite feeding on plant fluids, these insects often contain detectable quantities of plant DNA. Although inclusion of probes for bacterial *16S* in the original anchored hybrid probe kit yielded relatively high detection rates for chloroplast *16S*, the Illumina short reads also, in some cases, included DNA for various plant barcode genes as “by-catch”. Detection rates were generally only slightly higher for Typhlocybinae, which feed preferentially on parenchyma cell contents, compared to other groups of leafhoppers that feed preferentially on phloem or xylem. These results indicate that next-generation sequencing provides a powerful tool to investigate the specific association between individual insect and plant species.

## 1. Introduction

Associations between plants and insect herbivores, which have evolved over the past 400 million years, are a significant driver of terrestrial biodiversity [1]. Understanding how such associations evolved requires detailed knowledge of the specific associations between individual insect and plant species. Unfortunately, the host plants of many plant-feeding insect species remain unknown or uncertain. Many available host plant records have been acquired through casual observations by collectors rather than detailed study of feeding behavior. Thus, even many published host records may require further verification, depending on the source of the original information. The availability of improved DNA sequencing technologies and rapidly expanding online databases of DNA barcodes for many plants provides an alternate means for identifying the potential host plants of herbivorous insects. Field-collected plant-feeding insects may retain ingested plant tissue in the gut or attached to external surfaces of the body (e.g., pollen grains). Several previous studies have successfully amplified and sequenced plant barcode data from DNA extracted from insect bodies [2,3,4,5]. Until recently, most such studies have used standard PCR and Sanger sequencing approaches to amplify plant barcode gene fragments from samples of DNA extracted from the insect body. More recently, next-generation sequencing approaches have been used (e.g., [6]). The latter approach is particularly useful for identifying DNA from multiple plant species that may have been ingested by polyphagous insects [6].

Thus far, most such studies have focused on chewing herbivores such as beetles (Coleoptera) and caterpillars (Lepidoptera) that ingest leaf or stem material expected to contain large amounts of host plant DNA, or pollinators such as bees (Hymenoptera) and flies (Diptera) that retain pollen grains on external surfaces of the body [7]. Few attempts have been made to detect host plant DNA in sap-sucking insects (e.g., Hemiptera), many of which feed primarily on plant vascular fluids (phloem and/or xylem) and, therefore, may be less likely to ingest host plant DNA [8,9,10]. Thus far, the only studies of this type focusing on sap-sucking insects have used traditional PCR and Sanger sequencing [8,9,10,11].

Currently, four molecular markers are commonly used for plant species identification, including three plastid DNA regions, *rbcL*, *matK* and *trnH-psbA*, and the internal transcribed spacer (*ITS*) of nuclear ribosomal DNA [12,13]. In 2009, the Consortium for the Barcode of Life (CBOL) Plant Working Group proposed the combination of *rbcL* + *matK* as a “core barcode” for plant species [14]. The non-coding markers *trnH-psbA* (the intergenic region between *trnH* and *psbA*) and *ITS* (including *ITS1* and *ITS2*) are rapidly evolving, have strong discriminatory power and, therefore, have been widely used as supplementary barcodes for plant species identification and phylogeny reconstruction [15,16,17]. Compared to these barcodes, *16S rRNA*, a powerful and widely used marker for bacterial taxonomy [18], is less variable among chloroplast genomes of different plants. Consequently, it has been used primarily to construct plant phylogeny at deep levels [19,20].

The advent of next-generation sequencing technologies has yielded a wealth of new genome-scale DNA sequence data useful for a wide variety of potential applications. Of particular interest for this study are the Illumina short reads obtained through anchored hybrid enrichment [21]. Anchored hybrid enrichment aims to enrich a library of genomic DNA to increase the amount of DNA from specifically targeted regions of the genome relative to other regions using hybridization probes. In our case, probes were designed to target single-copy orthologous protein-coding genes. DNA from the original sample hybridizes with DNA probes, and this “captured” DNA is then submitted for high-throughput sequencing on the Illumina platform. In most cases, probes are designed to target the genomic regions of the species from which DNA is extracted. Nevertheless, inclusion of probes targeting other associated organisms (e.g., endosymbionts and pathogens) also works efficiently [22,23], although the quantity of DNA from these non-target organisms in such samples tends to be much lower. In addition to capturing the targeted loci, this method often yields DNA short reads from other regions of the genome, particularly genes present in high copy numbers such as mitochondrial and chloroplast DNA, referred to as “by-catch” [24]. Such “by-catch” is usually discarded as contaminant DNA during the assembly process. However, in some cases, it may provide additional useful data, for example, by revealing possible trophic associations with other organisms.

Here, we report the detection of plant barcode genes as well as chloroplast *16S rRNA* using anchored hybrid enrichment datasets generated from DNA extracted from sap-sucking insects (leafhoppers). Our results show that potential host plant DNA can sometimes be detected in “by-catch” sequence data from short read libraries obtained from such insects. We compare detection rates for DNA extracted from different parts of the insect body. We also compare detection rates for different subfamilies of leafhoppers, which either ingest contents of leaf parenchyma cells or feed on phloem sap.

## 2. Materials and Methods

We screened Illumina short read data previously generated from DNA extracted from insect bodies using anchored hybrid enrichment for a project focusing on the phylogenetics of leafhoppers (Hemiptera: Cicadellidae). Thus far, 1852 samples have been sequenced, including 1284 samples of Deltocephalinae, 178 Typhlocybinae, 68 Evacanthinae, 53 Eurymelinae, 47 Megophthalminae, 46 Cicadellinae, 42 Coelidiinae, 20 Mileewinae, 20 Tartessinae, 19 Ledrinae, 15 Iassinae, 12 Neocoelidiinae, 11 Ulopinae, 9 Nioniinae, 6 Bathysmatophorinae, 5 Aphrodinae, 5 Hylicinae, 4 Neobalinae, 4 Signoretiinae, 2 Errhomeninae and a single sample each of Phereurhininae and Portaninae. Insect specimens from which DNA was extracted were field collected into 95% ethanol over a 20-year period (1998–2018) and stored in −20 °C freezers (General Electric, Louisville, KY, USA) prior to being sorted, identified and placed in separate cryovials containing 95% ethanol (Decon Laboratories, King of Prussia, PA, USA). DNA extraction and anchored hybrid sequencing followed protocols described by Catanach and Dietrich (2018) [25] and Dietrich et al. (2017) [26]. In most cases, abdomens were removed from the body and used for DNA extraction. For small-sized insects (<5 mm), DNA was extracted from the whole body when multiple specimens were available. Legs were used for DNA extraction for a few large specimens (usually >2 cm). For some species, the abdomens of the only available specimen had previously been cleared in KOH solution; thus, DNA was extracted from the rest of the body. The anchored hybrid probe kit we used was a modified version of one used by Dietrich et al. (2017) [26] in a phylogenomic study of Membracoidea and included ~50 k probes mostly designed to capture sequence data from gene regions thought to be orthologous across this hemipteran superfamily [27]. Because the focus of the original study was on the phylogenetics of major lineages of leafhoppers and their primary bacterial endosymbionts, the probe kit did not include probes for plants or for insect mitochondrial genes. We screened the original Illumina short read data obtained from anchored hybrid enrichment of the leafhopper genomic DNA samples to test the feasibility of using this method to detect signatures of potential host plants of these insects.

### 2.1. Assemblies Based on Anchored Hybrid Enrichment Sequencing Data

Raw reads generated from anchored hybrid enrichment sequencing were processed using TrimmomaticPE [28] to remove adaptors and poor-quality data with a minimum length of 50, leading and trailing settings of 5, a slidingwindow setting of 4:15 and an Illuminaclip setting of 2:30:10. Cleaned reads were assessed for quality using FASTQC [29] and then assembled using ABySS 2.1.0 [30] with a k-mer length of 29 bp, 50 bp and 90 bp. Maximum length of blunt contigs to trim was set as 120 bp. Contigs with mean k-mer coverage less than three were removed.

### 2.2. Identification of Plant DNA

Four commonly used plant DNA barcodes, *rbcL*, *matK*, *trnH-psbA* and *ITS*, as well as chloroplast *16S rRNA,* were screened from the anchored hybrid assemblies of 1852 leafhopper samples. Reference sequences of the plant genes were downloaded from the National Center for Biotechnology Information (NCBI) Nucleotide Database (accession numbers are available in Appendix A), which represent nearly 70 orders of vascular plants. Nearly all the sequences of chloroplast genes, including tRNA (*trnH*), rRNA (*16S*) and protein-coding genes (*rbcL*, *matK* and *psbA*), were extracted from chloroplast genomes to yield full-length references. Due to the fast evolutionary rates of *trnH-psbA* (intron between *trnH* and *psbA*), *ITS1* and *ITS2*, the more conserved flanking genomic region *trnH*, *psbA* and *5.8S* rRNA (between *ITS1* and *ITS2*) were employed as query sequences instead. Using the nucleotide sequences of *16S rRNA*, *trnH* and *5.8S rRNA*, and the protein sequences of *rbcL*, *matK* and *psbA* as queries, BLASTN and TBLASTN searches were performed against the anchored hybrid assemblies with a cut-off E value of 10^−5^. Because our probe kit contains some probes targeting bacterial *16S rRNA*, a large number of sequences were screened as matching chloroplast *16S*. Therefore, the targeting sequences were then processed for a BLASTN search against a database containing the chloroplast *16S* query sequences and the bacterial *16S* sequences used for probe design. Sequences eligible for reciprocal best BLAST hits and with ≥800 bp matching length were considered as candidate genes for further identification. Only a few sequences matched the remaining plant markers; therefore, no second round of BLAST search was performed. The candidate sequences were trimmed according to the nucleotide sequences of the full-length reference genes (*16S*, *rbcL*, *matK*, *trnH-psbA* and *ITS1-5.8S-ITS2*), before being identified through BLASTN against the standard nucleotide databases in NCBI (default settings except a maximum of 250 target sequences were displayed). Retrieved sequences finally confirmed as plant genes are available in Appendix A.

## 3. Results and Discussion

A total of 139 plant sequences were retrieved using our anchored hybrid enrichment datasets, including 67 *16S rRNA*, 42 *matK*, 28 *rbcL* and 2 *ITS* (Figure 1A). Although a short *psbA* sequence was also identified, we failed to retrieve the flanking intron *trnH-psbA*. Approximately 6.7% of the samples (124 out of 1852) were shown to contain plant DNA. Compared to the DNA examples extracted from whole body or abdomen, the plant DNA detection rate is much higher in the samples extracted from the body without an abdomen (head and thorax) (Figure 1B). When comparing detection rates among different leafhopper subfamilies, we found that the highest detection rate occurred among Typhlocybinae, the microleafhoppers (Figure 1C). This may be because, unlike other leafhoppers, which feed on vascular fluids, species in this group pierce and suck the contents of parenchyma cells [31] which may have higher concentrations of plant DNA than phloem or xylem sap. Nevertheless, detection rates for other leafhopper groups thought to feed preferentially on xylem or phloem were, in some cases, only slightly lower than those for Typhlocybinae (Figure 1C).

Despite the lack of probes targeting these particular gene regions, we retrieved very short sequences for most of the plant barcode markers (Appendix A). Interestingly, nearly all the retrieved *rbcL* sequences hit some species of Cactaceae, while the majority of *matK* sequences match the genes sequenced from Rubiaceae samples, suggesting that some DNA probes included in our original anchored hybrid probe kit are similar to DNA sequences for these plant families. However, sequences obtained for the two core barcodes, ranging from 150 to ~600 bp, are too short to allow for confident species identification. One of the *ITS* sequences contains full-length *ITS1*, *5.8S rRNA* and *ITS2* (617 bp) and is identical to the *ITS* sequenced from *Lophostemon confertus*, a species native to Australia [32]. Interestingly, this sequence was retrieved from the sequence data of a microleafhopper that is endemic to Australia (*Dziwneono*, [33,34]). Although this marker has seldom been used solely to identify plant species, its strong discriminatory power, as well as the concordant geographic distributions between *L. confertus* and *Dziwneono*, suggests this association might be reliable.

In contrast, inclusion of probes for the bacterial *16S* rRNA gene in the original probe kit yielded a much higher detection rate for sequences identified as chloroplast *16S*, which far outnumber sequences for the plant barcode markers and are also much longer. Twenty-seven of these *16S* sequences are full length (≥1491 bp), and the remining sequences are longer than our cut-off of 800 bp. Unfortunately, chloroplast *16S* is a relatively slowly evolving locus that has mostly been used for plant phylogeny reconstruction rather than species identification. Thus, available data for this gene are not sufficient in most cases to identify the plant species with precision. Nevertheless, we found general agreement between the identities of the *16S* sequences obtained and known host records of some of the leafhopper species (Appendix A), including *Kahaono negrea* Dworakowska, 1972 feeding on *Eucalyptus* [33,35], *Copididonus hyalinipennis* (Stål, 1854) feeding on *Panicum* [36,37], *Zyginella pulchra* Löw, 1885 feeding on *Platanus* [38,39] and a new species of *Zyginama* feeding on *Quercus*. In addition, the top hit plant species of some sequences have the same geographic distribution as the leafhopper samples, indicating potential leafhopper–plant associations. For example, *Anzygina zealandica* (Myers, 1923) is distributed in Australia and New Zealand [40,41]. Host records of this species consist of a variety of plants, including many species of Fabaceae [34,42]. The sequence retrieved from data of *A. zealandica* is most similar to a *16S* of *Platylobium obtusangulum*, the common flat pea endemic to Australia [43]. *Variolosa meni* Cao & Zhang, 2013 was only recorded from Hainan Island, China [44]. The full-length *16S* retrieved from this species is identical to some species of *Machilus* and *Alseodaphnopsis*, including a species also distributed in Hainan (*Alseodaphnopsis hainanensis*, [45]). The *16S* screened from two species of *Neozygina* both hit *Prosopis*, a common shrub in the habitats where *Neozygina* occurs in the Southwestern USA [32,46].

Despite the high diversity of plant-feeding insects and their importance as actual or potential agricultural pests, reliable host plant data are lacking for the majority of species. Data on host associations obtained through field observations may or may not be reliable depending on whether the insect was (at minimum) observed feeding on a particular plant or (ideally) reared on a particular plant through an entire generation [47]. Many published records of host plant associations for herbivorous insects consist of collection records gleaned either from data labels attached to museum specimens or from the collector’s field notes. Such records generally do not reflect detailed observations of feeding behavior but, rather, indicate only that the insect was collected on a particular plant. Host records for the often highly mobile adults of phytophagous groups such as leafhoppers and planthoppers may require further validation if they are based only on field observations. Such insects frequently move from plant to plant and may be collected from non-food plants. Unfortunately, for many herbivorous insect species, a few (or even single) casual observations of this sort are the only data available on their possible host plant associations. Next-generation DNA sequencing methods provide a potentially powerful alternative means for identifying host plants using DNA extracted from preserved insect specimens. Our results suggest that this method can work even for sap-sucking insects not expected to ingest large quantities of host plant DNA. Although plant-specific DNA probes were lacking in the anchored hybrid probe kit used to generate the data used in this study, we were able to obtain at least partial sequences for several plant genes by screening by-catch data present among the short reads obtained from the Illumina platform.

The presence of bacterial *16S* probes in the anchored hybrid probe kit used for the original study yielded a much higher capture rate for chloroplast *16S* DNA than for other plant genes. Unfortunately, the nucleotide sequence of this particular gene is too conservative to be useful for species-level identification of plants, and the presence of plant *16S* DNA in many of the leafhopper samples screened suggests that these insects often retain plant DNA in their bodies, despite feeding primarily on plant fluids. Our ability to extract smaller fragments of other plant genes, including the main plant barcode loci, despite their lack of representation in the anchored hybrid probe kit used to generate the original data, is also promising. Furthermore, the proportion of samples containing plant DNA is likely to be higher than the results reported here, given the relatively stringent cut-off we applied when screening for chloroplast *16S*. Based on our results, we predict that data for these and other plant genes could be captured more reliably by incorporating DNA probes specific to these barcode genes into future anchored hybrid sequencing projects focused on plant-feeding insects, including the sap-sucking hemipteran insects used in our study.

To our knowledge, the design of such probes for capturing plant barcode DNA from samples containing mixtures of plant and animal DNA has not yet been attempted. Nevertheless, when comparing the available reference sequences across a variety of plant orders, we found conserved regions for all four plant barcode genes, either within the genes themselves (*rbcL* and *matK*) or in flanking regions (*trnH* and *psbA* next to *trnH-psbA*, *18S*, *5.8S* and *26S* upstream, between and downstream of *ITS1* and *ITS2*). The existence of such conserved genetic regions is promising and suggests that it should be possible to design anchored hybrid probes with high and universal efficiency for land plants.

The ability to detect host plant DNA by screening herbivorous insect DNA samples using this method presumably depends on a number of factors including the age and state of preservation of the original insect samples, whether or not the insect fed on a plant shortly before it was captured and preserved and the particular feeding mode (e.g., vascular fluid feeding versus feeding on parenchyma cell contents), factors that we were unable to control in our study. We were most successful at detecting plant DNA in Typhlocybinae, which feed preferentially on parenchyma cell contents, but we also captured plant DNA from substantial numbers of specimens and species belonging to leafhopper groups that feed preferentially either on xylem (Cicadellinae) or phloem sap (most other leafhopper groups). Detection rates were nearly identical for DNA obtained from whole-body extracts and extracts obtained only from the abdomen. The higher rate of detection for DNA extracted only from the head and thorax (minus the abdomen) is difficult to explain. This may be a sampling artifact since the number of specimens in the latter category was nearly an order of magnitude lower than for the other two categories (Figure 1B). Possibly, larger amounts of bacteria (e.g., endosymbionts) present in the leafhopper abdomen yielded a higher proportion of bacterial reads compared to plant DNA reads, which hindered detection of the latter in samples containing DNA from the abdomen. Further work is needed to optimize methods for detecting and identifying plant DNA in these insects.

## 4. Conclusions

Our ability to obtain potential host plant DNA sequences by screening by-catch data from anchored hybrid enrichment of DNA extracted from sap-sucking insects, despite the absence of plant-specific probes, indicates that such insects do ingest plant DNA in quantities sufficient to be detected by next-generation sequencing methods. Therefore, our results strongly suggest that next-generation sequencing methods represent potential powerful tools for identifying potential host plants of herbivorous insects with even small quantities of ingested plant DNA. Contrary to our expectations, capture rates for plant DNA were only slightly lower in leafhoppers that feed on vascular fluids (xylem or phloem) compared to those that feed on leaf parenchyma cell contents. Therefore, with appropriate enhancements, i.e., addition of plant-specific probes to anchored hybrid probe kits used for future projects, such methods show promise for greatly improving knowledge of insect–plant associations.

## Figures and Tables

**Figure 1 insects-12-00964-f001:**
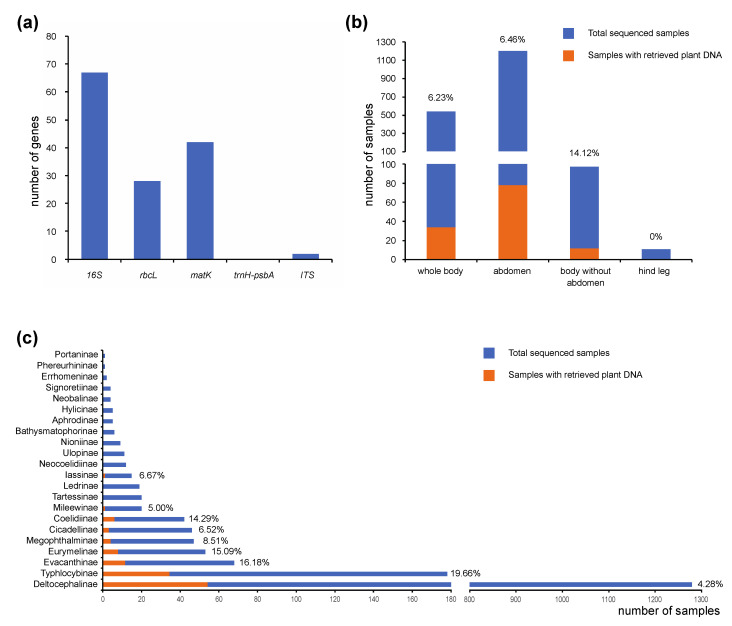
Statistics of “by-catch” plant DNA from leafhopper anchored hybrid enrichment datasets. (**a**) Number of different plant genes retrieved from anchored hybrid enrichment datasets. (**b**) Plant genes retrieved in the DNA samples extracted from different body parts. Plant DNA detection rates are indicated above bars. (**c**) Plant genes retrieved in each leafhopper subfamily. Plant DNA detection rates are indicated behind bars.

## Data Availability

Sequences of the identified plant genes can be found in Appendix A.

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
