# Peer review of "Identification of Potential Host Plants of Sap-Sucking Insects (Hemiptera: Cicadellidae) Using Anchored Hybrid By-Catch Data"

_insects, 2021, doi:10.3390/insects12110964_

Round 1

Reviewer 1 Report

Please refer to the annotated pdf file attached for more specific comments:

The authors present a proof-of-concept study that may lead to a very useful tool set for the identification of phytophagous insect host plants. The study was described in a coherent and easy to follow manner, though some general comments were made that lacked sources. 

Reviewer 2 Report

The present manuscript provides an important tool to increase our knowledge in insect-plant interactions. Using the NGS techniques to determine the potential host plants of insect species would help, not only to know deeply the trophic webs but also to develop integrated pest management strategies identifying potential vector species (as occurs with Xylella fastidiosa).  

I recommend this manuscript publication after some minor changes/ suggestions.

Line 106: Just curiosity, is there any difference among those samples from 2018 and the samples from 1998? The rates of plant DNA detection were similar?

Line 107: Does the ethanol and posterior cryopreservation (-20ºC) and then ethanol again affect to the DNA integrity and thus the later DNA sequencing analysis?

Line 108-109: You provided the refferences 25 and 26 for the DNA extraction (2 lines), compared to the rest of the M&M section it seems undercompensate. Due to the use of different insect parts in the plant DNA detection, Why dont you give some info about the insects dissection or the samples treatment?

Line 160-166: You are giving an explanation to your results and also you include a citation, these sentences should be in discussion. 

Lines 175-177; 182- 204:  the same than before, you are citting and discussing in the results section.

Given that "Insects" do not have strict formatting requirements, why don't you join the results and discussion sections? I consider that It will improve the manuscript making it more fluid for the readers allowing you to put more emphasis in each result (or relevant species), without changing the structure and citations that you currently have in the results section.

In the introduction (58-59) you comment that most of the studies are focused in carabids, why don't you compared your  species detected with those publications? 

Line 255-271: different font compared to the rest of the text.
